# MODEL EXTRACTION ATTACKS ON ARABIC BERT-BASED APIS

**Hassan A. H. Ali & Mohammed S. E. Eltahir**
Department of Electrical and Electronics Engineering
University of Khartoum
Khartoum, Sudan
{hassankarim8934,mseeltahir}@gmail.com

**Khalid N. Elmadani**
Department of Computer Science
University of Cape Town
Cape Town, South Africa
ahmkha009@myuct.ac.za

**Anas H. Showk**
Department of Electrical and Electronics Engineering
University of Khartoum
Khartoum, Sudan
anas.showk@uofk.edu

## ABSTRACT

In this paper, we study the feasibility of performing Model Extraction attacks on Arabic BERT-based APIs. In our experiments, we try to perform these attacks under different scenarios and observe the accuracy of the extracted model against the victim model. We then propose a method for protecting against these types of attacks by introducing noise, in the form of pre-training the victim model for more epochs on non-public generic data. Our results show that this strategy better secures the victim model from such attacks.

## 1 INTRODUCTION

**BERT** (Devlin et al., 2018) -Bidirectional Encoder Representations from Transformers- is a pre-trained language model whose architecture is composed of a stack of Transformer's Encoders Vaswani et al. (2017). The power of BERT extends beyond other languages; the architecture of BERT facilitated pre-training models on different languages like Arabic. These models include AraBERT (Antoun et al.), ArabicBERT (Safaya et al., 2020), SudaBERT (Elgezouli et al., 2021) and others.

Model Extraction is a process in which an adversary model with only query access to a victim model can mimic the behavior of that model without any knowledge about its internal parameters. It is not even required to query the victim model with real data and that was demonstrated to be successful through many NLP tasks (Krishna et al., 2019). (Keskar et al., 2020) demonstrates that the extracted knowledge from the victim model can be directly used with multilingual models in other languages without the need to perform the extraction for every language. Furthermore, adversarial queries that are fed to the extracted model to produce incorrect results can be transferred to the victim model (He et al., 2021).

In this paper, we explore model extraction on Arabic BERT-based models through different scenarios and propose a protection strategy for the victim model.

## 2 METHODOLOGY

In this paper, we perform two sets of experiments, the **Model Extraction** experiments and the **Protection** experiments. In each model extraction attempt, we fine-tune Arabic-BERT (Safaya et al., 2020) (which is the base model) with LABR dataset (Aly & Atiya, 2013) to create the victim model. Then the model extraction attack is conducted by passing different queries to the victim and using its prediction along with the queries to train the extracted model. The workflow of the model extraction attack is shown in figure 1.

We study different scenarios for extracting the victim model. Firstly, we assume that the attacker

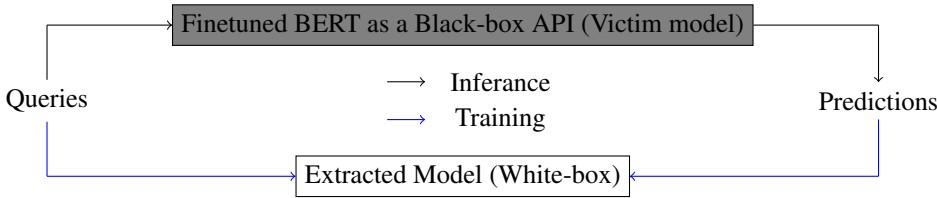

Figure 1: Model Extraction Attack performed by passing queries to victim model to get predictions that are used with the queries to train the extracted model.

| Model | #Q | Arabic-BERT | | SudaBERT | |
|---|---|---|---|---|---|
| | | Accuracy | Loyalty | Accuracy | Loyalty |
| **Victim** | | 85.21% | - | 83.97% | - |
| **Arabic-BERT** | | | | | |
| $D_A = D_V$ | | 86.22% | 93.11% | 85.47% | 89.42% |
| $D_A \neq D_V$ | 1x | 83.03% | 90.55% | 82.63% | 88.35% |
| $D_A \neq D_V$ | 5x | 84.52% | 91.45% | - | - |
| **AraBERT** | | | | | |
| $D_A = D_V$ | | 87.32% | 89.42% | 86.42% | 87.12% |
| $D_A \neq D_V$ | 1x | 86.02% | 89.77% | 84.62% | 89.25% |

Table 1: The results for extracting Arabic-BERT and SudaBERT victim models using Arabic-BERT and AraBERT. Loyalty represents how good is the extracted model in mimicking the behavior of the victim. #Q is the number of queries.

knows the base model. In this case, we experiment with the following scenarios of extracting the victim model using: 1) the same training examples as queries ($D_A = D_V$), 2) the same amount of queries (1x) but from a different dataset ($D_A \neq D_V$), 3) more queries (5x) from another dataset ($D_A \neq D_V$).

The other assumption is that the adversarial doesn't know the base model. Here we conduct two experiments for extracting the victim using AraBERT (Antoun et al.). In one experiment, we used the same training examples as queries ($D_A = D_V$). While in the other experiment, we used the same amount of queries (1x) but from a different dataset ($D_A \neq D_V$). In the cases of using a different dataset for extraction ($D_A \neq D_V$), we sample the queries from the BRAD dataset (Elnagar & Einea, 2016).

In our protection strategy, we suggest pre-training the victim for more epochs on more non-public unlabeled generic data before it is fine-tuned. We used SudaBERT (Elgezouli et al., 2021), which is Arabic-BERT that is trained for more epochs on $7M$ sentences of Sudanese dialect. We think that introducing this noise would change the model parameters enough so that it is different from the base model, which would make it difficult for the adversarial model to extract.

## 3 RESULTS

Table 1 shows the results of our experiments. We assess the performance of the extracted model in two measures, accuracy and loyalty. The accuracy measures how accurate the model is against the test data of the LABR dataset. Loyalty is calculated using the predictions of the victim model as a reference for testing. It represents how well the extracted model is able to mimic the behavior of the victim.

The first row in the table represents the results of fine-tuning the victim models on LABR dataset. The following two blocks represent the results of extracting the victim using Arabic-BERT and AraBERT. For the model extraction experiments, results show that the loyalty is always higher when extracting the victim using the same base mode (Arabic-BERT). While the accuracy is higher when using the victim's training data as queries ($D_A = D_V$). In other words, loyalty depends more on the pre-trained extractor, while accuracy relies more on the queries used for the extraction. For the protection experiments, the model that was pre-trained on additional unlabeled data (SudaBERT) was

harder to extract since the extracted model has less loyalty, but this comes in cost of the SudaBERT victim model has less accuracy than the Arabic-BERT victim model.

## 4 CONCLUSION

In this paper, we study extracting Arabic BERT-based models. Results show that these attacks were successful regardless of the base model and data that the adversary uses to extract the victim model. We also propose a protection strategy for the victim model and use it to reduce the loyalty of the best extracted model.

### URM STATEMENT

The authors acknowledge that at least one key author of this work meets the URM criteria of ICLR 2023 Tiny Papers Track.

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

## A  EXPERIMENTAL SETUP

### A.1  DATASETS

- **LABR** dataset (Aly & Atiya, 2013): is a collection of book reviews that has 11760 rows for training, and 2900+ for testing. (we used the balanced version of this dataset).
- **BRAD** dataset (Elnagar & Einea, 2016): is a collection of book reviews that has a total of 510,000+ rows. We take three different portions of this dataset , the first one has the same size as LABR dataset ($D_A \neq D_V$) 1x . the second portion has five times the size of LABR dataset ($D_A \neq D_V$) 5x and the last one has 2000 rows that used for testing the royalty of the extracted model. These datasets are used in the conditions where the LABR dataset which victim model what fine-tuned on assumed to be not available.

### A.2  HYPERPARAMETERS

For all models, we use a batch size of 16, a learning rate of $5e - 5$, and 3 epochs for finetuning. All experiments were done using the Huggingface Transformers library [1]. The optimizer used was the Adam optimizer.

---

[1] `https://huggingface.co/docs/transformers`

