# OpenReview forum: "Model Extraction Attacks on Arabic BERT-Based APIs"
_ICLR.cc/2023/TinyPapers — Submitted to Tiny Papers @ ICLR 2023_

### Official Review · Reviewer_iTcm · 2023-04-01

**Confidence:** 3

**Summary Of Contributions:**

The paper explores model extraction attacks on BERT models pre-trained on Arabics such as AraBERT and SudaBERT. The paper also proposes a method for protecting model extraction attacks through adding the noise in form of pre-training the victim model on more epochs.

**Rating:**

Clear, Correct, and Reproducible (CCR): a submission which meets the reviewing criteria

**Strengths And Weaknesses:**

**Strengths:**

- The paper is written well, clearly and it is concisely. It presents the problem in an understandable way. Good abstract and literature review.
- The conducted experiments are presented well. Nice use of figures and tables.
- The paper provides a good amount of details of used datasets.
- The paper follows ICLR paper requirements.

**Weaknesses**

- There are not enough details of the architecture used during pre-training(example: size of BERT model, number of encoders, dimension of the model).

**Suggested Changes:**

- It’s nice that you provided some hyperparameters in the experimental setup/appendix. The paper would also benefit from including the details about the architecture used and other hyperparameters that are not reported(like the type of optimizer used).
- Minor changes: correct CONCULSATION(section name).

---

### Meta-Review · Area_Chair_pv4s · 2023-04-08

**Recommendation:** Invite to present
**Confidence:** 4

**Metareview:**

This paper studies model extraction attacks on Arabic BERT-based APIs. The paper is well-written. The problem is clearly defined, and the experiments are convincing.


**Summary:**

This paper explores the model extraction attack and defense. The work is presented well. The weakness (minor) is the lack of detailed implementation of the approach.

**Reason For Not Giving A Higher Recommendation:**

N/A

**Reason For Not Giving A Lower Recommendation:**

Overall, this is solid work. It studies an interesting and clearly defined problem. The proposed technique is sound and effective. The results are insightful.

---

### Decision · Program_Chairs · 2023-04-09

**Decision:**

Invite to present

**Comment:**

Please reformat your abstract so it shows correctly in OpenReview.